# Effect of Sign-Alternating Cyclic Polarisation and Hydrogen Uptake on the Localised Corrosion of X70 Pipeline Steel in Near-Neutral Solutions

**Alevtina Rybkina [1], Natalia Gladkikh [1], Andrey Marshakov [1], Maxim Petrunin [1] and Andrei Nazarov [2],***

[1]  Frumkin Institute of Physical Chemistry and Electrochemistry, Russian Academy of Sciences, 119071 Moscow, Russia; aa_rybkina@mail.ru (A.R.); fuchsia32@bk.ru (N.G.); mar@ipc.rssi.ru (A.M.); mmvp@bk.ru (M.P.)

[2]  French Corrosion Institute, 29200 Brest, France

*   Correspondence: andrej.nazarov@institut-corrosion.fr; Tel.: +33-(0)-2-98-05-15-52

**Abstract:** The effect of sign-alternating cycling polarisation (SACP) on the localised corrosion of X70pipeline steel in solutions of various compositions was studied. Localised corrosion of steel at anodic potentials was accelerated with an increase in the duration of the cathodic half-cycle, in the presence of a promoter of hydrogen absorption in aqueous electrolyte, and with an increase in the concentrations of chloride and bicarbonate ions. It was pointed out that the corrosion rate is determined by the amount of hydrogen absorbed by the steel. A quantitative indicator to determine the intensity of localised corrosion under SACP was suggested.

**Keywords:** pipeline steel; sign-alternating polarisation; pitting corrosion; soil electrolyte; hydrogen uptake

---

## 1. Introduction

Corrosion of carbon and low-alloy steels under the effect of stray current is among the most hazardous types of corrosion damage of metal structures operating in soils or in natural waters. As a rule, the effect of stray current from a DC source can be different from the effect of current induced by an AC source (e.g., high-voltage power lines) [1]. On the other hand, stray current from DC sources vary with time. If cathodic protection of a structure is used, it results in oscillations in the polarisation potential (*E*), and the instantaneous value of *E* can be either more positive or more negative than the free corrosion potential of steel [2,3]. Hence, the effect of stray current from DC and AC sources leads to alternating-sign polarisation of a metal in a structure, which differs mainly in the frequency of potential oscillations.

It was initially supposed that the corrosion rate of steel is determined by dissolution during the anodic half-cycle of the alternating current. The effect of the cathodic on the anodic reaction was not taken into account, and corrosion was considered as a case of Faraday rectification [4–6]. In some cases, a qualitative relationship between the amplitude of the AC potential and the rate of AC corrosion was found. However, no quantitative agreement was observed for many corrosion systems [7–11]. In addition, the AC corrosion of steels is of a mainly localised nature, and corrosion in pits can occur at sufficiently negative potentials such as are applied for cathodic protection [4–16]. These features of AC corrosion of steels have been explained by a significant increase in the pH of the near-electrode solution [1,17]. In alkaline media, passivation of the metal occurs in the AC anodic half-period. In the negative half-wave period, the passive film is reduced to Fe (II) hydroxide/oxide species and a new passive film grows again during the next anodic cycle. When the newly formed passive film is reduced,

the amount of Fe (II) compound increases. As a result, a certain amount of the metal is oxidised during each cycle, which results in a considerable mass loss of the steel structure due to corrosion.

A similar process scheme was suggested to explain the effect of rectangular potential pulses on steel corrosion, which are supposed to simulate the alternating-sign zone of stray currents from a DC source [18–20]. According to [20], at the cathodic protection potential (−0.65 V vs. saturated hydrogen electrode, SHE), steel is in the passive state due to the high pH. A shift of potential in the anodic direction gradually activates the metal surface due to a decrease in near-surface pH. A cycling of the potential only in the anodic direction from E = −0.65 V (SHE) accelerates the steel corrosion to a smaller extent comparing to a 'two-sided' potential shift both in the negative and positive directions around the cathodic protection potential [20].

Electrode activation-passivation caused by a variation in near-surface pH is not the only reason for localised steel corrosion under alternating polarisation [11]. It has been shown that under cathodic polarisation, 'pit-like' defects could form on the surface of pipeline steel [21,22]. This was explained as a result of hydrogen charging and the formation of high hydrogen pressure at metal/ non-metallic inclusion interfaces [22]. As the potential is shifted in the anodic direction, intense dissolution of the hydrogen-saturated metal layer inside the pits occurs. If the potential step is cycled, a corrosion location grows and is transformed to a micro-crack in mechanically stressed metal [23]. Dai et al. demonstrated that an increase in the frequency of the rectangular potential pulses in a simulated soil electrolyte increases the coverage of the surface of X100 pipeline steel with pits [24]. This was explained by a shift in the potential of the double electric layer in a positive direction. However, the effect of solution composition on the intensity of localised corrosion of steel was not considered. The effect of hydrogen adsorption in the cathodic cycle can also be important for the corrosion of steel at the rest potential or during anodic polarisation. Thus, the objective of this work was to study the effect of sign-alternating cycling polarisation (SACP) and the composition of a pH-neutral solution on the localised corrosion of pipeline steel.

## 2. Materials and Methods

The studies were carried out using samples of X70 pipeline steel machined from a pipe (manufactured by KhTZ Du Enterprise, Kharkov, Ukraine, pipe diameter 1420 mm and thickness 18.7 mm) along the centreline at a distance of 120 mm from the longitudinal weld. The chemical composition of the steel is presented in Table 1. Samples of discs with a working surface area of 1.93 $cm^2$ were used. Samples were polished using emery paper and finished with a diamond paste of grit size down to 0.5 μm. The samples were degreased for 25 min in an ultrasonic bath using ethanol/toluene 1:1.

**Table 1.** The chemical composition of X70 pipeline steel (wt.%).

| C | Si | Mn | P | S | Cr | Ni | Cu | Al | Ti |
|---|---|---|---|---|---|---|---|---|---|
| 0.115 | 0.34 | 1.63 | 0.021 | 0.003 | 0.04 | 0.02 | 0.007 | 0.030 | 0.07 |

The following working electrolytes were used: borate buffer (BB) with the composition: 0.4 M $H_3BO_3$ + 5.5 mM $Na_2B_4O_7$, pH 6.7; a mixture of BB and solution (NS4) with the composition 1.64 mM KCl + 5.75 mM $NaHCO_3$ + 1.23 mM $CaCl_2 \cdot 2H_2O$ + 0.74 mM $MgSO_4 \cdot 7H_2O$ [25]; and the solutions with addition of 0.01 M thiourea (TU), 0.1 M and 0.6 M NaCl, 0.1 M $NaHCO_3$, and IFKhAN-29 inhibitor (1 g/L). The compositions of the solutions are shown in Table 2. All solutions were prepared from reagents of chemical grade and de-ionised water. The experiments were carried out with free access to oxygen at room temperature of 22 ± 2 °C. This indicates electrochemical studies, the samples were subjected to cathodic pre-treatment for 15 min at a potential of −0.65 V (SHE).

**Table 2.** Specimen surface characteristics (total area $S_p$, density $\rho$, mean diameter $d$ and maximum depth $h$ of pits) after 72 cycles of SACP ($-1$ V $\leftrightarrow$ $-0.3$ V vs. SHE) with duration of anodic polarization $\tau_a$ 3 min and various durations of cathodic polarization $\tau_c$ in different solutions.

| Electrolyte Composition | $\tau_c$, min | $S_p \times 10^3$, mm$^2$/cm$^2$ | $\rho$, pits/mm$^2$ | $d$, μm | $h$, μm |
|---|---|---|---|---|---|
| BB | 10 | 0.006 | 2 | 2 | 1 |
| BB + NS4 | 0 | 0.73 | 22 | 6.5 | 3 |
| BB + NS4 | 10 | 1.13 | 32 | 6.7 | 4 |
| BB + NS4 | 60 | 1.66 | 30 | 8.4 | 4 |
| BB + NS4 + TU | 0 | 1.63 | 23 | 9.5 | 5 |
| BB + NS4 + TU | 10 | 4.72 | 68 | 9.4 | 7.5 |
| BB + NS4 + TU | 60 | 17.90 | 100 | 15.1 | 9 |
| 0.1 M NaCl | 10 | 3.56 | 60 | 8.7 | 10 |
| BB + 0.1 M NaCl | 10 | 2.72 | 60 | 7.6 | 6.5 |
| BB + 0.6 M NaCl | 10 | 5.17 | 39 | 13 | 7.5 |
| BB + 0.1 M NaHCO$_3$ | 10 | 0.84 | 17.5 | 7.8 | 2 |
| BB + 0.1 M NaHCO$_3$ + TU | 10 | 2.27 | 43 | 8.2 | 3 |
| BB + NS4 + TU + IFKhAN-29 | 60 | 1.76 | 22 | 10.1 | 4 |

The potential was cycled from a cathodic value ($E_c = -1$ V) to an anodic value ($E_a = -0.3$ V) using an IPC Pro-MF potentiostat ("Volta", Saint Petersburg, Russian Federation). A standard three-electrode electrochemical cell described in [26] was used. The potentials are reported versus SHE. The duration of the cathodic SACP period ($\tau_c$) was 10 or 60 min and the duration of the anodic period ($\tau_a$) 3 min. The number of potential cycles in all the experiments was the same (72 cycles). At the end of an experiment, the sample was withdrawn from the electrochemical cell and the corrosion products removed with hydrochloric acid/methylamine (1:1). The sample was washed in distilled water, degreased with ethanol and the surface was photographed using a Biomed PR-3 optical microscope (manufactured by Biomed Service LLC, Moscow, Russia) and an AC-300 digital video camera recorder (Amoyca, Ho Chi Minh City, Vietnam) mounted on the ocular. The camera resolution was 2048 × 1536 pixels. Data from the camera were evaluated with Scope Photo 3.0 software ("Scope Tec", Munich, Germany) to determine the number of pits ($N$) and the surface area of the sample occupied by each pit ($S_i$). Initially, $N$ and $S_i$ were determined from surface images with low magnification and refined using images with higher magnification (Figure 1). The pit depth ($h$) was determined using a Neophot-2 metallographic microscope. The mean maximum value of $h$ was determined from the values observed for the three deepest pits.

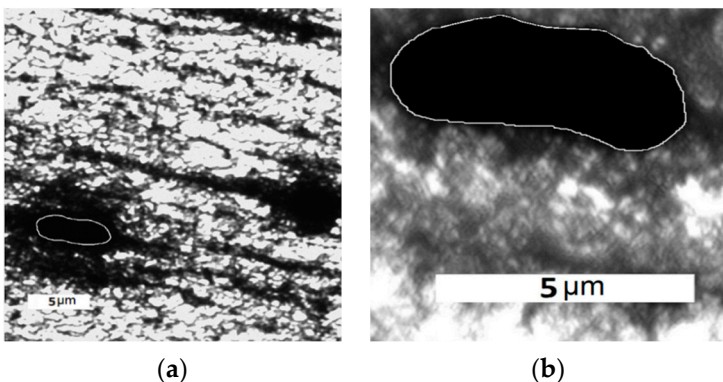

(a)    (b)

**Figure 1.** Images the sample surface after SACP at $\tau_c = 10$ min in 0.1 M NaCl solution. Magnification is (**a**) 5×; (**b**) 20×.

The total surface area of a sample covered by pits was calculated as:

$$S_P = \sum S_i / S \tag{1}$$

The coverage of the sample surface with defects was determined as:

$$p = N/S \tag{2}$$

where $S$ is the visible area of the sample (1.00 mm$^2$).

Assuming that pits have a hemispherical form, the mean defect diameter was calculated:

$$d = \sum d_i/N \tag{3}$$

where $d_i = \sqrt{S_i/\pi}$.

The difference (tolerance) in the values of S$_p$, $\rho$, and $d$ obtained on different parts (1 mm$^2$) of a working surface area did not exceed 15% of the average values. The rate of hydrogen permeation into the metal was measured using a steel hydrogen pickup sensor whose sensing principle was based on the Devanathan-Stachurski setup [27]. The sensor membrane was made of steel foil with a thickness of 100 μm, and the working area was 33.1 cm$^2$. A Pd film was deposited cathodically on the membrane exit side for 100 s at a constant current density of 25 mA/cm$^2$. The Pd plating solution contained 25 g/L PdCl$_2$ and 20 g/L NH$_4$Cl. The solution pH was adjusted to 8.5 by adding the required amount of NH$_4$OH. Prior to the tests, the Pd-plated membranes were degassed in a dry-air desiccator at room temperature for 2 days. The diffusing part of the cell was filled with 0.1 M NaOH and the permeation current was measured at a potential of 0.2 V (SHE). The electrochemical experiments for hydrogen sensing and optical microscopy were carried out using the same setup.

## 3. Results and Discussion

Based on images of the sample surfaces similar to those shown in Figure 1, the total area occupied by pits (S$_p$, mm$^2$), their density ($\rho$, mm$^{-2}$) and mean diameter ($d$, μm) were determined (Table 2). The same table shows the maximum depth of pits, ($h$, μm) and data obtained at a constant anodic potential $E = -0.3$ V ($\tau_c = 0$) and different durations of the cathodic half-cycle ($\tau_c = 10$ min and $\tau_c = 60$ min). In all setups, the total duration of anodic polarisation cycle was 216 min.

Figure 2 shows the variation in the total area (a), density (b), an average diameter (c) of pits versus time of cathodic part of the cycle ($\tau_s$) in NS4 solution in borate buffer (BB + NS4) as the background and with the addition of $10^{-2}$ M thiourea (BB + NS4 + TU). The same setup of SACP was used either with or without addition of tiourea. The addition to the solution of a promoter of hydrogen absorption (thiourea) significantly increases the total area and density of pits (Figure 2a,b), and at $\tau_c = 60$ min, the diameter of pits (Figure 2c). The maximum pit depth also increases markedly with increasing $\tau_c$ in the solution containing thiourea (Table 2). As can be seen, the total area of pits increases upon SACP and the greatest effect is observed with the longest cathodic period ($\tau_c = 60$ min). Localised corrosion occurs more intensely as the duration $\tau_c$ increases. Figure 2 shows that the increase of the duration of cathodic polarization increases the corrosion of the steel with or without the addition of thiourea in borate electrolyte. It was determined that in all cases thiourea accelerates the corrosion of the steel comparing with reference electrolyte. This allows us to conclude that hydrogen charging of steel during the cathodic period leads to its accelerated localised dissolution during the anodic period [23,28].

It can be supposed that pits appear as a result of hydrogen charging of the pipeline steel and the formation of locations of high hydrogen pressure, predominantly at the metal/non-metallic inclusion interfaces [22,23]. Hydrogen absorbed by the metal accelerates the dissolution of steel in pH-neutral media at anodic potentials [29]. Due to the dissolution of the hydrogen-saturated metal layer inside the pits, the diameter and depth of these should increase.

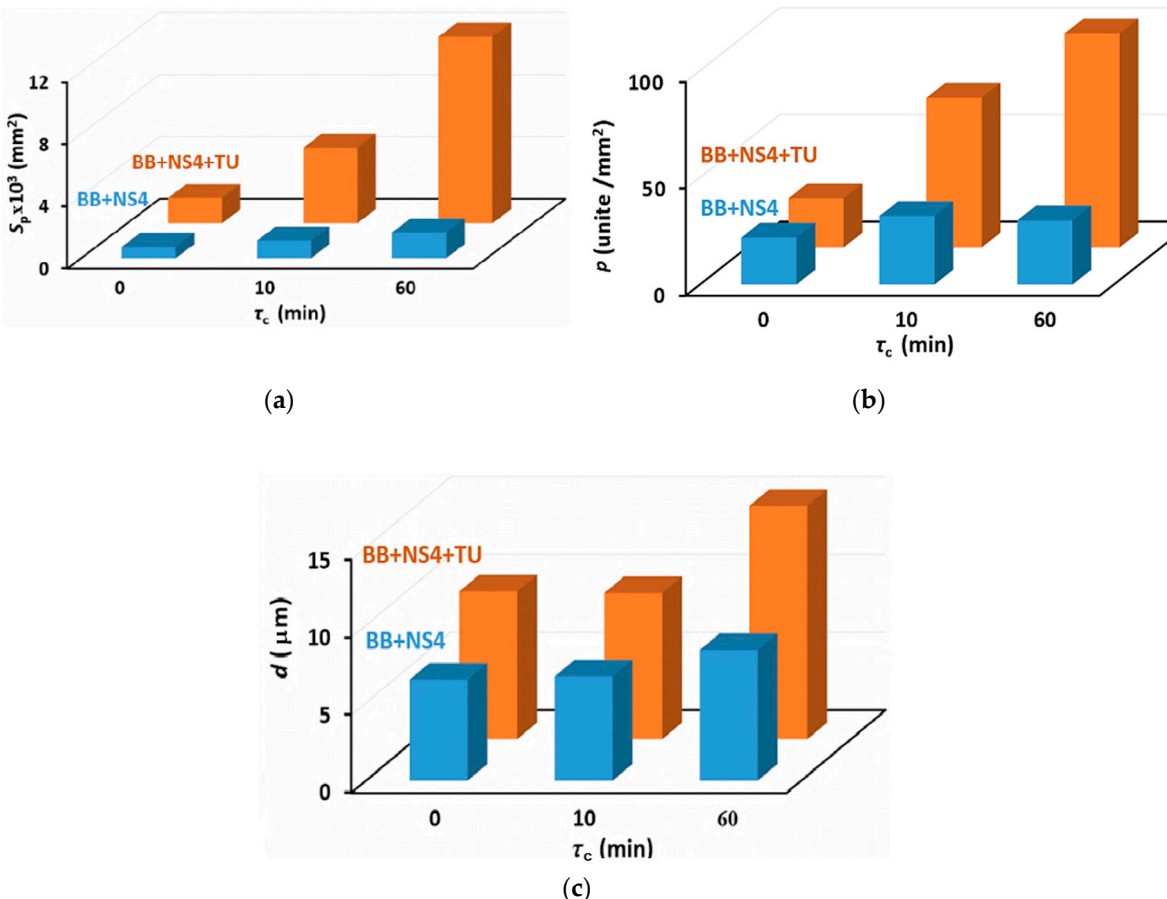

**Figure 2.** The effect of addition of $10^{-2}$ M of thiourea on the total pit area $S_p$ (**a**); pit density on the sample surface (**b**) and their average diameter d (**c**) versus duration of the cathodic period of the cycle in SACP. The setup consists from 72 cycles, the duration of anodic period of one cycle 3 min.

As follows from Table 2, the intensity of localised corrosion increases with increasing concentration of chloride ions [$Cl^-$] in the solution. Figure 3 (curves 1) shows the impact of chloride concentration in BB electrolyte on the total area, density, diameter, and depth of pits after cathodic polarisation for 10 min and the following anodic cycle. As can be seen, the values of $S_p$, $\rho$, $d$, and $h$ increase abruptly on transition from the pure borate buffer to the BB + NS4 mixture, which can be a result of NS4 solution containing not only chloride but also bicarbonate and sulphate ions. However, with an increase in [$Cl^-$] to 0.1 M, the area, density and depth of pits increase significantly compared to the values observed in the BB + NS4 solution (Figure 3a,b) containing a low concentration of aggressive ions. In BB + 0.6 M NaCl solution, the density of pits decreases (Figure 3b) but their diameters increase (Figure 3c). As a result, the total pit area increases (Figure 3a). Apparently, the density of pits decreases because they are merged together.

In non-buffered 0.1 M NaCl electrolyte, the values of $S_p$, $\rho$, $d$ and $h$ (Figure 3, points 2) are much greater than in borate buffered electrolyte or in the same solution with NS4. This confirms that chloride ions accelerate the localised corrosion of steel during cathodic-anodic cycling. Borate buffer does not affect the density of pits but slows down their growth (Figure 3b). As a result, the values of $S_p$, $d$ and $h$ in the BB + 0.1 M NaCl solution are smaller than in a pure chloride solution of the same concentration (Figure 3a,c,d).

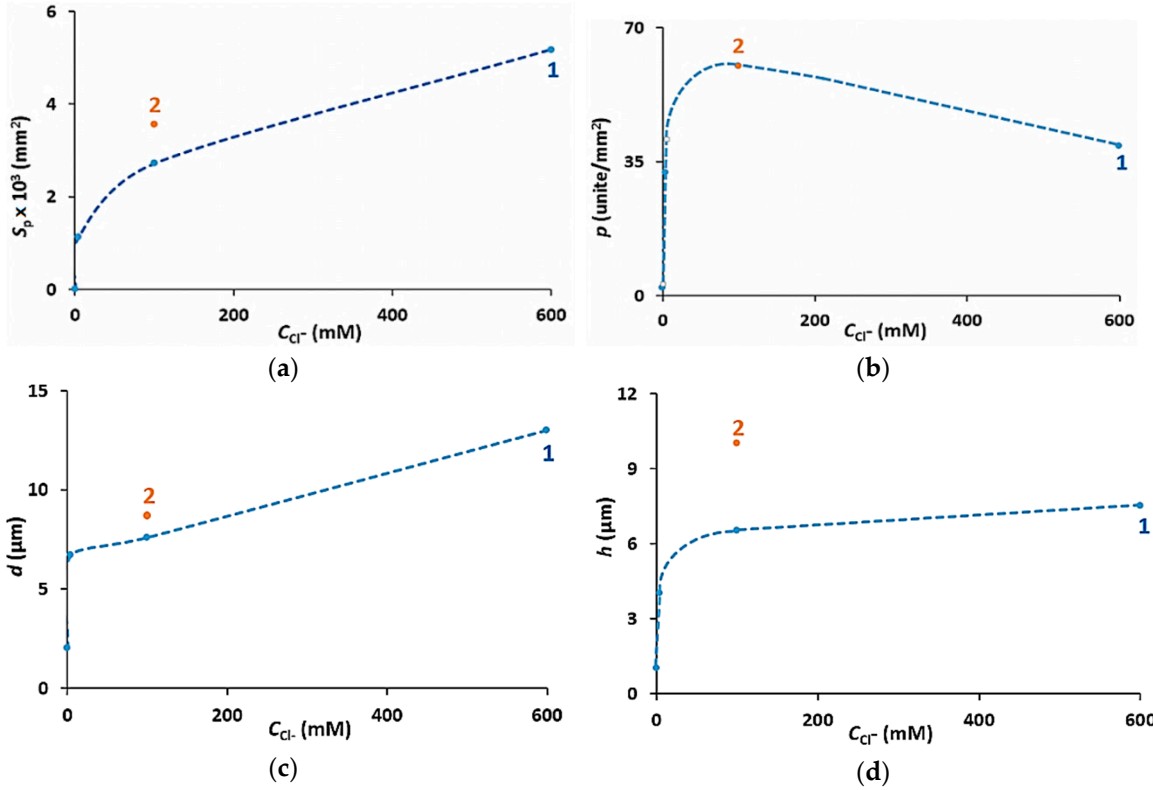

**Figure 3.** Variation in the total area $S_p$ (**a**); density $\rho$ (**b**); mean diameter $d$ (**c**); and maximum pit depth $h$ (**d**) as a function of the concentration of chloride ions (curves 1) in borate buffer electrolyte. Point 2—non-buffered 0.1 M NaCl aqueous solution. The setup consists from 72 cycles of SACP (the potential switched between $-1$ V vs. SHE (duration 10 min) and $-0.3$ V vs. SHE (duration 3 min).

Corrosive electrolytes in soils often contain carbonate and bicarbonate ions, and one of the objectives was to determine the effect of these ions on pitting corrosion. Table 2 shows that the addition of 0.1 M $NaHCO_3$ affects the localised corrosion of X70 steel. Addition of bicarbonate to borate electrolyte significantly increases both the total area $S_p$ and the density of the pits $\rho$. The mean pit diameter $d$ increases almost four-fold. However, bicarbonate affects the characteristics of localised corrosion to a smaller extent than does chloride. Thus, the ratios of $S_p$ and $\rho$ values in BB + 0.1 M NaCl and BB + 0.1M $NaHCO_3$ solutions are 3.3 and 3.4, respectively (Table 2). The pit diameter in these solutions is approximately the same (Table 2); i.e., chloride favours the initiation of more numerous pits.

Cathodic polarisation can lead to damage of steel constructions due to hydrogen-assisted steel cracking. The addition of a hydrogen uptake promoter such as thiourea to BB electrolytes containing 0.1 M $NaHCO_3$ significantly activates the localised corrosion, increasing the density of pits, but variation in their diameter is not observed. On the other hand, localised corrosion occurs more intensely in the BB + NS4 + TU solution that in BB + 0.1 M $NaHCO_3$ + TU (Table 2). This also confirms that chloride ions have a stronger effect on pit nucleation and growth relatively bicarbonate ions. The pit depth in BB + 0.1 M $NaHCO_3$ and BB + 0.1 M $NaHCO_3$ + TU solutions is insignificant: 2 and 3 μm respectively (Table 2). Hence, a sufficiently high concentration of bicarbonate inhibits the growth in the depth. This is in line with the results for the corrosion of pipeline steel in simulated soil electrolytes [30].

The addition of IFKhAN-29 corrosion inhibitor to the BB + NS4 + TU solution inhibits local steel dissolution during cathodic-anodic cycling ($\tau_c$ = 60 min, Table 2). In this case, the density of pits decreases significantly (by a factor of almost 5). Hence, the inhibitor primarily affects the pit nucleation process. The growth of pits is also inhibited; the diameters and depths decrease by factors of 1.5 and 2.2, respectively (Table 2). Thus, it is possible to point out that the intensity of the localised corrosion, which is characterised by the $S_p$, $\rho$, $d$ and $h$ values, increases with an increase in the duration of the

cathodic period of the cycle and in the presence of a promoter of hydrogen absorption. It depends also on the ionic composition of the electrolyte, such as by the addition of chloride or bicarbonate anions.

The results show that the hydrogen charging of steel promotes both the nucleation of pits during the cathodic period and their growth during the anodic period of the cycle, which is in line with the literature [22,23,28]. It can be expected that the rate of localised corrosion is determined by the amount of hydrogen absorbed by the steel at the cathodic potential. It is important to find a quantitative integral parameter that would determine the occurrence of localised corrosion of pipeline steel due to sign-alternating polarisation. To determine the hydrogen flux into the steel during cathodic polarisation, an electrochemical technique to measure the hydrogen penetration current ($i_p$) across a membrane was applied. Table 3 shows $i_p$ as function of the solution composition during steel cathodic polarisation.

**Table 3.** The stationary rate of hydrogen penetration ($i_p$) into steel at $E = -1$ V (SHE), the stationary density of anodic current ($i_{a,st}$) at $E = -0.3$ V (SHE) and the anodic current density in 0.09 s after switching the potential ($i_0$) under SACP with $\tau_c = 10$ min in solutions of various compositions.

| Electrolyte Composition | $i_p$, A/cm$^2$ | $i_{a,st}$, mA/cm$^2$ | $i_0$, mA/cm$^2$ |
|---|---|---|---|
| BB | 6 | 0.020 | 0.17 |
| BB + NS4 | 10 | 0.020 | 0.20 |
| BB + NS4 + TU | 90 | 0.116 | 0.57 |
| 0.1 M NaCl | 14.5 | 0.041 | 1.85 |
| BB + 0.1 M NaCl | 18 | 0.100 | 1.27 |
| BB + 0.6 M NaCl | 38 | 0.082 | 3.8 |
| BB + 0.1 M NaHCO$_3$ | 18 | 0.067 | 0.44 |
| BB + 0.1 M NaHCO$_3$ + TU | 27.6 | 0.113 | 1.05 |

Figure 4 correlates the total area of the corroded surface during SACP (time of cathodic polarisation 10 min) versus the hydrogen permeation current as a function of electrolyte composition. Figure 4 brings together the data obtained in BB, BB + NS4, BB + NS4 + TU solutions (curve 1), BB, BB + 0.1 M NaCl, BB + 0.6 M NaCl solutions (curve 2), and BB, BB + 0.1 M NaHCO$_3$, BB + 0.1 M NaHCO$_3$ + TU solutions (curve 3). It is obvious that hydrogen entry to the steel during cathodic polarisation accelerates the pitting dissolution that is taking place during the anodic period of the cycle.

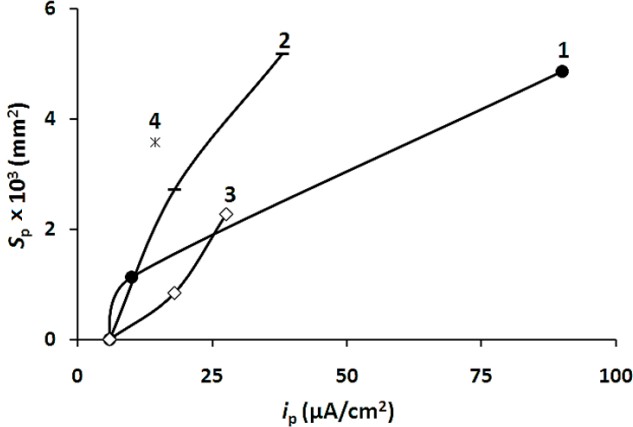

**Figure 4.** Correlation between the total pits area $S_p$ (duration of cathodic part of SACP 10 min and anodic part 3 min, total 72 cycles) and the hydrogen penetration current $i_p$ in solutions: 1 (·) BB, BB + NS4, BB + NS4 + TU; 2 (-) BB, BB + 0.1 M NaCl, BB + 0.6 M NaCl; 3 (◊) BB, BB + 0.1 M NaHCO$_3$, BB + 0.1 M NaHCO$_3$ + TU; point 4 (∗) is non-buffered 0.1 M NaCl aqueous electrolyte.

As can be seen in Figure 4, an increase in the concentrations of chloride and bicarbonate results in an increase in the rate of hydrogen incorporation into the steel. At the similar hydrogen permeation current ($i_p$), an increased area of pitting corrosion ($S_p$) was observed in chloride solutions, while a

lower corroded area was observed in bicarbonate-containing solutions. Thus, the corrosion rate is not influenced only by the hydrogen uptake but also the kind of electrolyte species.

Current transients obtained after switching cathodic and anodic potentials can give additional information about the mechanism of corrosion of the pipeline steel at SACP. Figure 5 shows the transients of the anodic (solid lines) and cathodic (dashed lines) currents in logarithmic coordinates obtained during potential cycling for some electrolytes and shows significantly different results. One can see that at the initial period (up to about 0.1 s) after switching of the potential, the cathodic current ($i_c$) is slightly higher than the anodic current ($i_a$), but the difference increases with time. The slope of the log $i_c$-log $\tau$ curves decreases with time and the cathodic current tends to reach a constant value. The slope of the log $i_a$-log $\tau$ curves remains nearly constant for some time (the duration of this section on the curves depends on the solution composition). Subsequently, the anodic current stabilises or increases with time. A increase in the slope of log $i_a$-log $\tau$ curves could be due to the formation of a primary passive film on the metal. Similar anodic current transients were observed on iron in BB solution [31]. The formation of a layer of iron oxides on the electrode surface was shown in solution by method of quartz resonator [32]. However, at $E = -0.3$ V, a continuous barrier layer of iron oxide/hydroxide compounds is not formed [32]. Hence, the current at the end of the anodic SACP half-cycle ($i_{a,st}$) relates to the uniform dissolution of the metal and localised dissolution in the pits formed at the cathodic potential.

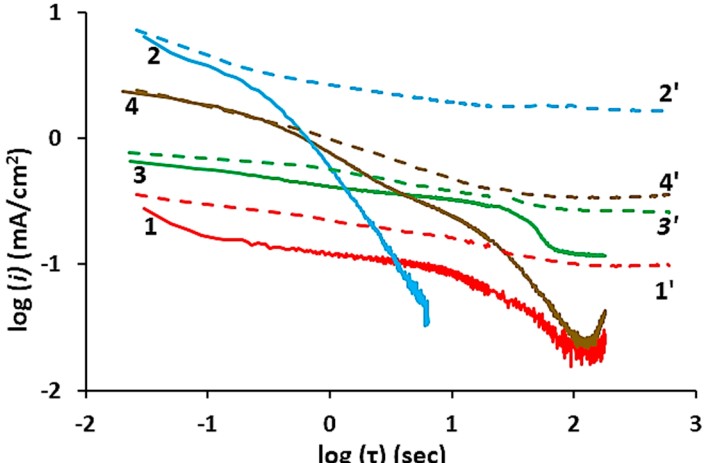

**Figure 5.** Transients of the anodic currents at $-0.3$ V vs. SHE (1–4) and cathodic currents at $-1$ V vs. SHE (1'–4') in solutions: (1,1') BB; (2,2') BB + 0.6 M NaCl; (3,3') BB + NS4 + TU; (4,4') 0.1 M NaCl.

It is important to discuss the anodic current transients at the beginning of the passivation process in more details. Anodic current transients (Figure 5) were presented in $i_a$-$\tau^{-0.5}$ coordinates in Figure 6a. One can see that, in a certain time range, the current transients obtained in BB + 0.6 M NaCl (Figure 6a, curve 2) and 0.1 M NaCl solutions (Figure 6a, curve 4) are linearized in Cottrell coordinates, and the extrapolated linear section passes through the origin of coordinates at $\tau \to \infty$. This is an indication that nonstationary diffusion in semi-infinite approximation is the limiting step of the anodic process. The anodic process of hydrogen-charged steel can involve the ionisation of atoms of iron, alloying elements and the hydrogen atoms absorbed by the alloy. It has been reliably established that the active dissolution of iron under stationary conditions is limited by the kinetic of electron transfer [33]; therefore, the nonstationary anodic process cannot be limited by the diffusion of $Fe^{2+}$ ions in the liquid phase. In principle, the dissolution of alloying components of steel might be limited by their solid-phase diffusion in iron. However, due to small values of the diffusion coefficients of alloying components in the metal matrix, this possibility seems unlikely, since the linear portions of the $i_a$-$\tau^{-0.5}$ curves can be rather long, for example, up to 20 s in 0.1 M NaCl solution (Figure 6b, anodic curve).

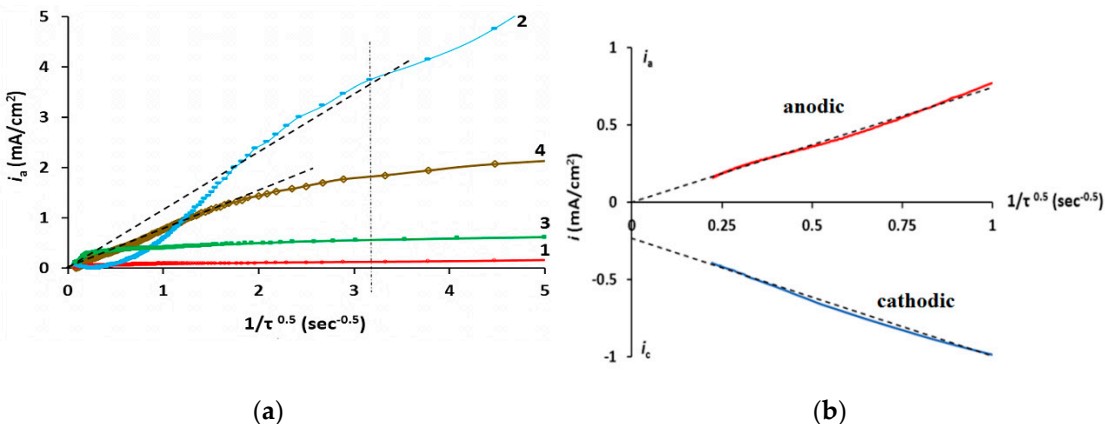

**Figure 6.** (**a**) Transients of anodic current at −0.3 V (SHE) in solutions: (1) BB; (2) BB + 0.6 M NaCl; (3) BB + NS4 + TU; (4) 0.1 M NaCl. (**b**) Initial parts of anodic (−0.3 V vs. SHE) and cathodic (−1 V vs. SHE) current transients in 0.1 M NaCl aqueous electrolyte.

The cathodic current transients obtained in BB + 0.6 M NaCl and 0.1 M NaCl solutions are also linearized in the $i_c$-$\tau^{-0.5}$ coordinates in a certain time range. For example, Figure 6b shows the linear section of the cathodic current transient obtained in 0.1 M NaCl solution. The linear sections of the $i_a$-$\tau^{-0.5}$ and $i_c$-$\tau^{-0.5}$ plots (Figure 6b) are observed in the same time range; their slopes show opposite signs and are nearly equal. This indicates that the anodic and cathodic current transients are related to the same process, such as nonstationary solid-phase diffusion of hydrogen atoms in the metal [33]. However, the cathodic evolution and entry of hydrogen on iron and steels consist of a number of reactions. The reactions of the evolution of hydrogen gas proceed in parallel with the incorporation of hydrogen into the metal. Therefore, the extrapolated linear section of the $i_c$-$\tau^{-0.5}$ dependence (Figure 6b, cathodic curve) does not pass through the origin but shifts down on the Y-ordinate axis. This shift in cathodic current should be close to the rate of hydrogen gas evolution. In fact, it is nearly equal to the stationary cathodic current and to the rate of cathodic reaction because the hydrogen penetration current into the metal is relatively small (if the electrode thickness is sufficient). In addition to hydrogen oxidation, the anodic process in hydrogen-charged steel should include a parallel reaction of metal ionisation. However, extrapolation of the linear section of the $i_a$-$\tau^{-0.5}$ curve to the origin (Figure 6b, anodic curve) shows that in the present experimental setup, the ionisation rate of the metal in 0.1 M NaCl solution is small compared to the rate of oxidation of absorbed hydrogen.

The initial section of the $i_a$-$\tau^{-0.5}$ curves deviates from a linear plot at lower currents (Figure 6a). As a rule, this effect results from the kinetic limitations of the reaction rate. For example, hydrogen extraction from Pd and its alloys over a short time is limited by the process of hydrogen desorption from the metal phase [34]. It can be assumed that the weak dependence of the current on $\tau^{-0.5}$ observed in our experiments in the initial period (Figure 6a) is also due to the kinetic limitations of the interface reaction of hydrogen desorption. This is in line with the data on anodic current transients obtained on an iron electrode in BB that is described by the equation of diffusion-phase-boundary kinetics of hydrogen extraction from the metal [35,36]. In fact, for different reasons, a quantitative interpretation of the current transients obtained on steel is difficult. The hydrogen charging of steel under SACP should occur locally in the corrosion defects. Additionally, the gradient of hydrogen concentration in the steel is nonlinear because hydrogen is mainly absorbed by the surface layer of a metal [37,38]. A primary passivating film is formed during SACP, which should inhibit both iron dissolution and hydrogen effusion from the metal (the diffusion coefficient of hydrogen in the iron oxide layer is lower by several orders of magnitude than in the metal [37,38]). Nevertheless, it can be pointed out that in the initial period the anodic current is primarily determined by the rate of hydrogen extraction (oxidation) from the steel.

It can be expected that the hydrogen extraction from local corrosion events, such as the bottom and the 'banks' of pits would also accelerate dissolution of the metal due to defectiveness of the subsurface layer of the alloy. A similar effect is observed for selective dissolution of a number of alloys; the dissolution of the electropositive alloy component is activated due to dissolution of the electronegative component [39,40]. Thus, it can be supposed that a relationship should exist between the intensity of localised steel corrosion under SACP and the current value at the beginning of the anodic half-cycle (Figure 5). Figure 7a shows the correlation between the total area of pits and the anodic current taken at $\tau = 0.09$s. The value of $\tau = 0.09$s was taken arbitrarily, but the curve (Figure 7a) does not change significantly for the currents measured at a lower initial time. The $S_p$-$i_0$ plot (Figure 7a) is nonlinear with a coefficient of regression $R^2$ of 0.63. The point TU obtained in the BB + NS4 + TU solution containing the promoter of hydrogen absorption is far from the $S_p$-$i_0$ curve. The reason for the intensive localised corrosion of steel in this solution is a result of a high rate of hydrogen penetration into the metal ($i_p$, see Table 3). Therefore, for an understanding of the mechanism of pitting corrosion of X70 steel in SACP cycles, the current of hydrogen penetration ($i_p$) has also been taken into account.

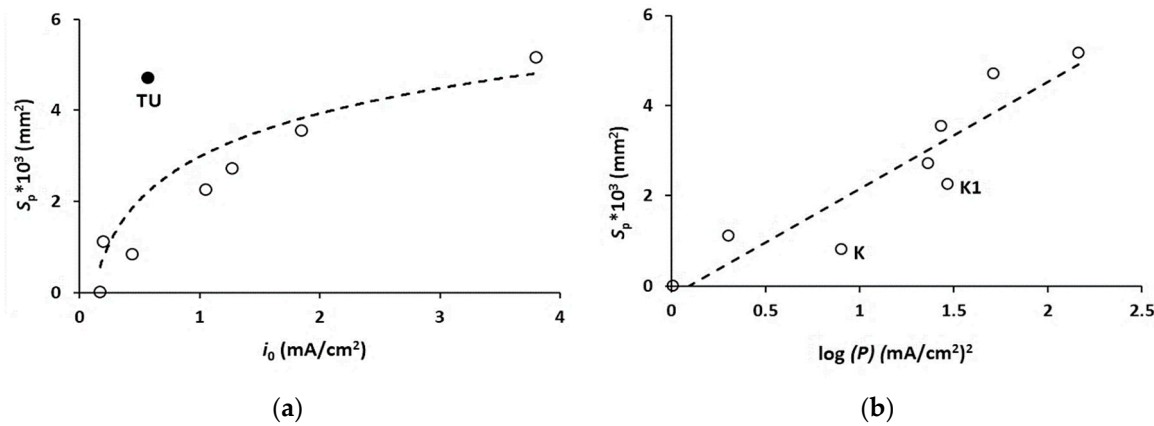

**(a)**          **(b)**

**Figure 7.** (**a**) Correlations between total area of the pits $S_p$ and anodic current ($i_0$) at the beginning of anodic cycle; (**b**) total area of the pits $S_p$ vs. parameter $P$ (Equation (4)) related to the intensity of steel corrosion during SACP cycles.

Figure 7b shows the relationship between surface area containing pits ($S_p$) and the product ($P$) of anodic current ($i_0$) at the beginning of an anodic cycle and the current ($i_p$) of hydrogen penetration across the membrane (Equation (4)). $P$ is the empirical parameter determining the rate (surface area) of pitting corrosion that shows the most promising correlation. Considering the nonlinear nature of $S_p$ vs. $P$, the plot is presented in semi-logarithmic coordinates:

$$i_0 \times i_p = P \tag{4}$$

$$S_p = 2.37 \log P - 0.22 \tag{5}$$

The resulting regression (Equation (5)) relates to all points presented in Figure 7b with a coefficient of determination $R^2$ of 0.85. The points K and K1 (Figure 7b), corresponding to solutions BB + 0.1 M NaHCO$_3$ and BB + 0.1 M NaHCO$_3$ + TU, are located below the linear regression line. Apparently, in solutions with a high content of bicarbonate ions, insoluble FeCO$_3$ can precipitate during steel dissolution and inhibit the growth of corrosion defects at the bottom of pits [30]. Thus, it can be supposed that the intensity of localised corrosion of pipeline steel under SACP (Figure 7b) depends both on the amount of hydrogen absorbed during the cathodic half-period and on the rate of hydrogen extraction during the anodic half-period.

An integral quantitative indicator $P$ (Equation (4)), determining the intensity of localised corrosion of steel during SACP, contains the stationary hydrogen penetration current through a steel membrane

(at the potential of the cathodic period) and the anodic current measured at the beginning of the anodic period of SACP. However, this indicator does not take into account less-controlling factors occurring during cyclic alternating polarisation, such as the formation of a layer of insoluble corrosion products.

## 4. Conclusions

(1) The intensity of localized corrosion of pipeline steel X70 increases with an increase of duration of the cathodic period of sign-alternating cycling polarization. This effect was found either with or without the addition of a promoter of hydrogen absorption (thiourea) to background electrolyte. The thiourea increases the total area and density of pits significantly. Thus, effective hydrogen charging of steel during the cathodic period accelerates localized dissolution during the anodic period of SACP.

(2) The corrosion of pipeline steel during anodic period of SACP increases with an increase of concentration of chloride and bicarbonate ions in solution. Bicarbonate ions influence on the rate of localized corrosion to a smaller extent than chloride ions. Chloride ions accelerate the nucleation of pits and increase their maximum depth.

(3) The IFKhAN-29 corrosion inhibitor affects the nucleation of the pits and hinders the localised corrosion of pipeline steel in the anodic period of an SACP cycle. The inhibitor reduces the surface density of pits by a factor of nearly five.

(4) The nature of current transients in SACP as a result of switching of the potential was investigated. It was shown that in the initial period, cathodic and anodic currents are determined by the rates of hydrogen absorption and extraction from the metal.

(5) An integral quantitative indicator of the intensity of localised corrosion of X70 steel in different solutions during SACP was proposed with a satisfactory coefficient of determination ($R^2 = 0.85$). This is the product of two currents: the current of hydrogen penetration through a steel membrane in the cathodic period and the anodic current of hydrogen extraction measured at the beginning of the anodic period of the SACP.

**Author Contributions:** A.R.: methodology, investigation, writing—review & editing, visualization; N.G.: methodology, software, investigation; A.M.: writing-original draft preparation, project administration; M.P.: data curation, funding acquisition; A.N.: writing—original draft preparation, supervision. All authors have read and agreed to the published version of the manuscript.

**Funding:** This research was funded by the Ministry of Science and High Education of the Russian Federation, No. AAAA-A20-120012390029-7.

**Conflicts of Interest:** The authors declare no conflict of interest.

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
