# Peer review of "Effect of Sign-Alternating Cyclic Polarisation and Hydrogen Uptake on the Localised Corrosion of X70 Pipeline Steel in Near-Neutral Solutions"

_metals, doi:10.3390/met10020245_

Round 1

Reviewer 1 Report

The paper provides a sound report on the effect of stray polarization and hydrogen uptake on the localized corrosion of pipeline steel with a balanced presentation of new reports and review of previous achievements. The paper is suitable for the journal, and it is recommended for acceptance subject to minor revision.

A) Corrections:

line 49: use underscript for CuSO4

line 62: al.

Figure 2 (b): Use ρ for the Y axis

Figure 3 (b): Use ρ for the Y axis

line 251: "proceed"

Figure 7 (a): Use "TU" instead of "UT"

Figure 7 (b): Use "log" in the X axis, for consistency with the other occurrences of the logarithmic function in the manuscript

B) Revise various obscured/incomplete sentences:

line 54: "around the potential cathodic protection"

line 164: "varies insignificantly". Variation is not observed.

lines 185-186: the method to detect hydrogen penetration is not described in the experimental section

lines 215-216: "and then increases sharply". Not observable in Figure 5, at least as a sharp increase. There is increase only for curve 4 among 5 curves.

lines 271-272: "the hydrogen adsorbs the surface layer of a metal"

line 326: "with the satisfied coefficient of determination"

lines 375 and 377: journal abbreviation: Corros. Eng. Sci. Technol.

line 382: What is the use of "//"? Title missing

lines 386 and 418: journal abbreviation for Corrosion (NACE): "Corrosion"

line 395: What is the use of "//"?

line 410: "for studying"

C) Recommendations:

Use of a common reference for the potential values in the paper. The CSE reference electrode is used for the values reported in the Introduction section, whereas the SHE is used in the rest of the paper. It is recommended to refer to SHE the values in the Introduction. Use the name "pipeline steel" in all occurrences, instead of switching between this name and "pipe steel".

Author Response

Dear Mr. Reviewer! We are very grateful to you for attention and patience! Thank you very much for all your comments that allowed us to correct the flaws! Responses to your comments are highlighted in yellow in the manuscript.

Comments and Suggestions for Authors

The paper provides a sound report on the effect of stray polarization and hydrogen uptake on the localized corrosion of pipeline steel with a balanced presentation of new reports and a review of previous achievements. The paper is suitable for the journal, and it is recommended for acceptance subject to minor revision.

A) Corrections: The article has been proofread by Proof-Reading-Service.com

line 49: use under script for CuSO4

We listened to your recommendations, so all values were given relative to a standard hydrogen electrode. Previous values have been deleted.

line 62: al.

Agree. Changed.

Dai at al.

Figure 2 (b): Use ρ for the Y-axis

Agree. Changed.

Figure 3 (b): Use ρ for the Y-axis

Agree. Changed.

line 251: "proceed"

Agree. Changed.

The reactions of the evolution of the gas hydrogen proceed parallel with the hydrogen incorporation into the metal.

Figure 7 (a): Use "TU" instead of "UT"

Agree. Changed.

Figure 7 (b): Use "log" in the X-axis, for consistency with the other occurrences of the logarithmic function in the manuscript

Agree. Changed.

B) Revise various obscured/incomplete sentences:

line 54: "around the potential cathodic protection"

Agree. Changed.

A cycling of the potential only in the anodic direction from E = -0.85 V (CSE) accelerates the steel corrosion to a smaller extent comparing to a “two-sided” potential shift both in the negative and positive directions around the potential of the cathodic protection [20].

line 164: "varies insignificantly". Variation is not observed.

Agree. Changed.

The addition of hydrogen uptake promoter such as thiourea to BB electrolyte containing 0.1 M NaHCO3 significantly activates the localized corrosion that increases the density of pits, whereas their diameter variation is not observed.

lines 185-186: the method to detect hydrogen penetration is not described in the experimental section

Agree. Added this information in part 2 (page3)

The rate of hydrogen penetration into the metal was measured using a steel hydrogen pickup sensor that sensing principle is similar to the Devanathan-Stachurski setup [27]. The sensor membrane was made of steel foil with a thickness of 100 μm, the working area was 33.1 cm2. A palladium film was cathodically deposited on the membrane exit side for 100 s at a constant current density of 25 mA/cm2. The palladium plating solution contained 25 g/l PdCl2 and 20 g/l NH4Cl. The solution pH was adjusted to 8.5 by adding the required amount of NH4OH. Before the tests, the Pd-plated membranes were degassed at room temperature for no less than 2 days. The diffusing part of the cell was filled with 0.1 M NaOH and the membrane was polarised at a potential of 0.2 V. The electrochemical experiments for hydrogen sensing and optical microscopy were carried out using the same setup.

lines 215-216: "and then increases sharply". Not observable in Figure 5, at least as a sharp increase. There is an increase only for curve 4 among 5 curves.

Agree. We deleted the part of this phrase.

The slope of the log ia - log τ curves remains nearly constant for some time (the duration of this section on the curves depends on the solution composition).

lines 271-272: "the hydrogen adsorbs the surface layer of metal"

Agree. Changed.

Additionally, the gradient of hydrogen concentration in the steel is nonlinear because hydrogen is mainly absorbed by the surface layer of a metal

lines 375 and 377: journal abbreviation: Corros. Eng. Sci. Technol.

Thank you! Changed.

Huo, Y.; Tan, M. Y. Measuring and understanding the critical duration and amplitude of anodic transients. Corros. Eng. Sci. Technol. 2017, 1, 1 – 8. Huo, Y.; Tan, M. Y. Localized corrosion of cathodically protected pipeline steel under the effects of cyclic potential transients. Corros. Eng. Sci. Technol. 2018, 5. 348 – 354.

line 382: What is the use of "//"? Title missing

Thank you for your attention! We did not see that we missed the title. Changed.

Nenasheva, T. A.; Marshakov, А. I.; Kasatkina, I. V. The formation of local foci of corrosion of pipe steel under cyclic alternating polarization. : Mat. Prot. 2015, 5, 9 – 17 (in Russian).

lines 386 and 418: journal abbreviation for Corrosion (NACE): "Corrosion"

Agree. Changed.

Parkins, R. N.; Blanchard, W. K.; Delanty, B. S. Transgranular stress corrosion cracking of high-pressure pipelines in contact with solutions of near neutral pH. Corrosion. 1994, 5, 394 – 408. Qin, Z.; Demko, B.; Noel, J.; Shoesmith, D.; King, F.; Worthingham, R.; Keith, K; Localized dissolution of mill scale-covered pipeline steel surfaces. Corrosion. 2004, 10, 906 – 914.

line 395: What is the use of "//"?

"//" error. Deleted.

Rybkina, A. A.; Kasatkina, I. V.; Gladkikh, N. A.; Petrunin, M. A.; Marshakov, А. I. Experimental determination of local corrosion damage development rate on surface of pipe steels in model soil electrolytes. Corros.: Mat. Prot. (in Russian). 2019, 3, 1 – 8.

line 410: "for studying"

Agree. Changed.

Zakroczymski, T. Adaptation of the electrochemical permeation technique for studying entry, transport and trapping of hydrogen in metals. Electrochim. Acta. 2006, 11, 2261 – 2266. C) Recommendations:

Use of a common reference for the potential values in the paper. The CSE reference electrode is used for the values reported in the Introduction section, whereas the SHE is used in the rest of the paper. It is recommended to refer to SHE the values in the Introduction.

Agree. Changed.

According to [20], at the cathodic protection potential (-0.65 V vs. saturated hydrogen electrode, SHE), steel is in the passive state due to a high pH. Shift of the potential in anodic direction gradually activates the metal surface due to a decrease in near-surface pH. A cycling of the potential only in the anodic direction from E = -0.65 V (SHE) accelerates the steel corrosion to a smaller extent comparing to a “two-sided” potential shift both in the negative and positive directions around the potential of the cathodic protection [20].

Use the name "pipeline steel" in all occurrences, instead of switching between this name and "pipe steel". 

Agree. We made changes in all text.

Reviewer 2 Report

The article is interesting enough to be publishable, however it needs to correct the English used and small misleading errors

Author Response

The article is interesting enough to be publishable, however it needs to correct the English used and small misleading errors

Submission Date

12 December 2019

Date of this review

02 Jan 2020 22:25:09

Dear Mr. Reviewer! We are very grateful to you for attention and patience! Thank you very much for all your comments that allowed us to correct the flaws! Responses to your comments are highlighted in yellow in the manuscript.

The article has been proofread by Proof-Reading-Service.com

Reviewer 3 Report

The paper need a revision of the language. The style is the most concerning and some phrases need to be corrected like degreasing when the authors mean cleaning, or at all instead of et al..

However, I feel the authors have failed to clearly present the effect of solutin composition on the intensity of the localised corrosion as is described in the introduction. The use of confusing and unappealing graphs with cryptic figure descriptions further obscures the results. The authors seem to be confused themselves for example "Figure 4 shows the logarithmic dependence (dashed line, 195 R2 = 0.47)", figure 4 does not contain dashed lines.

The authors should present the results in a clearer way, the they should improve the figures.

When the issue of presentation is addressed (results and figures) the paper will be for a proper revision and hopefully publishing, as the thematic is interesting.

Author Response

Dear Mr. Reviewer! We are very grateful to you for your patience and understanding! Thank you very much. No doubt the revision was helpful to improve the article.

Comments and Suggestions for Authors

The paper titled “Effect of sign-alternating cyclic polarization and hydrogen uptake on the localized corrosion of X70 pipeline steel in near-neutral solutions” investigated the effect of sign-alternating cycling polarization (SACP) on the localized corrosion of X70 in solutions with various compositions. They found that the localized corrosion rate of steel is accelerated with increasing cathodic half-cycle duration, the hydrogen update and the concentrations of chloride and bicarbonate ions. And they also proposed a quantitative indicator to determine the intensity of localized corrosion of X70 under SACP condition. The results are interesting, but the manuscript in the current version cannot be accepted for publication. I recommend that this article be revised and re-reviewed. Some comments as following:

About the electrochemical tests, is the solution aerated or deaerated? Because oxygen has a significant effect on the cathodic reaction. What’s the solution temperature when the electrochemical tests were conducted?

Agree. The experiments were carried out with free access to oxygen at room temperature of 22 ± 2°C. These data were added to the manuscript.

Before the electrochemical tests, did you apply cathodic current on the specimen surface for depolarisation or not?

Before conducting electrochemical studies, the sample was subjected to cathodic pre-treatment for 15 minutes at a potential of -0.65 V (SHE). This information was added to the text.

Figure 1, the quality of the images is not good enough, it is better to replace these figures with high-quality images. And it is better to add some SEM pictures for the surface and cross-sectional morphologies of these samples.

Thanks for your comment!  The quality was upgraded using imaging software treatment. The explanation is added to the text.  As we continue our research, we plan to further use SEM microscopy.

Figures 2, 3, and 4, the error bar of all data points should be added.

All experiments were carried out in a single copy. The statistical treatment was added out using different optical images of the pits.

Line 247-249, “this indicates that … such as nonstationary solid-phase diffusion of hydrogen atoms in the metal”, some references are needed.

Agree. The reference was added.

Figure 7, “UT” should be “TU”.

Agree. Changed.

What’s the meaning of the parameter P? why did the author use such a parameter to identify the localized corrosion of X70 steel in SACP condition?

Pitting corrosion increased after the addition of thiourea (promoter of hydrogenation) and as a result of increasing the time of hydrogen absorption. P is just an empirical parameter determining the rate (surface area) of pitting corrosion as a function of hydrogen concentration and the rate the anodic current.  Correlation analyses showed that a combination of these two parameters (equation 4) related to hydrogen effusion (the rate of hydrogen oxidation and the hydrogen concentration) mainly determines the steel corrosion.

The phrase was added to the text. “P is an empirical factor determining the rate (surface area) of pitting corrosion that shows the most promising coefficient of determination.”

Considering all the problems mentioned here above, some parts of this manuscript should be revised. And more evidence is needed to support the statements of this manuscript. I recommend that the manuscript should be revised and reviewed again.

The article has been proofread by Proof-Reading-Service.com

Reviewer 4 Report

The paper titled “Effect of sign-alternating cyclic polarization and hydrogen uptake on the localized corrosion of X70 pipeline steel in near-neutral solutions” investigated the effect of sign-alternating cycling polarization (SACP) on the localized corrosion of X70 in solutions with various compositions. They found that the localized corrosion rate of steel is accelerated with increasing cathodic half-cycle duration, the hydrogen update and the concentrations of chloride and bicarbonate ions. And they also proposed a quantitative indicator to determine the intensity of localized corrosion of X70 under SACP condition. The results are interesting, but the manuscript in the current version cannot be accepted for publication. I recommend that this article be revised and re-reviewed. Some comments as following:

About the electrochemical tests, is the solution aerated or deaerated? Because oxygen has a significant effect on the cathodic reaction. What’s the solution temperature when the electrochemical tests were conducted? Before the electrochemical tests, did you apply cathodic current on the specimen surface for depolarisation or not? 1, the quality of the images is not good enough, it is better to replace these figures with high-quality images. And it is better to add some SEM pictures for the surface and cross-sectional morphologies of these samples. 2, 3, and 4, the error bar of all data points should be added. Line 247-249, “this indicates that … such as nonstationary solid-phase diffusion of hydrogen atoms in the metal”, some references are needed. 7, “UT” should be “TU”. What’s the meaning of the parameter P? why did the author use such a parameter to identify the localized corrosion of X70 steel in SACP condition?

Considering all the problems mentioned here above, some parts of this manuscript should be revised. And more evidence is needed to support the statements of this manuscript. I recommend that the manuscript should be revised and reviewed again.

Author Response

Dear Mr. Reviewer! We are very grateful to you for your work! Thank you very much. No doubt the revision was helpful to improve the article. The English has been proofread by Proof-Reading-Service.com

Round 2

Reviewer 3 Report

The authors have improved the language.

The author have made an overall minor revision, mainly based on the language. The results especially the implications they make are not presented clearly enough. In fact they have not addressed this issue at all.

For example:

The intensity of localised corrosion of pipeline steelX70 increases with an increase in the duration of cathodic polarisation in SACP cycles and with the addition of a promoter of hydrogen  absorption (thiourea). The thiourea affects the total area and density of pits most significantly.

Does this mean that the localised corrosion is increased by the duration of the c. pol. cycle or only when it's duration is increased in the presence of thiourea. The authors should also clearly state that the thiourea increases the total area density, a great reduction in corrosion would also mean a significant affect. The authors have not made any attempt to improve this issue.

The main issue is still the overall quality of presentation.

The "non-metal inclusion" should be changed to non-metallic inclusion as is the proper name. 

Figure quality has improved, but the descriptions under figures should be self explanatory and should make sense on their own, the description under figure 3 "Points 2 - see explanation in the text" is not acceptable. just write non-buffered 0.1 M NaCl, the condition of the testing should also be written under the figures, or in the graph, there is a lot of space, since there is only one curve.

Author Response

The author have made an overall minor revision, mainly based on the language. The results especially the implications they make are not presented clearly enough. In fact they have not addressed this issue at all.

Dear Mr Reviewer, thank you very much for your work, time and great contribution to the article quality. In fact, we respond to all your comments and did corresponding changes in the manuscript during the first run of review. However, only after the second run, we understood what we have to do. The article was double-checked in order to find the phrases with a possible miss understanding. These phrases were clarified. The captions to the Figures were rewritten to add information about the setups and conditions of particular experiments.  We hope that at present the article will be accepted for publication.

For example:

The intensity of localized corrosion of pipeline steelX70 increases with an increase in the duration of cathodic polarisation in SACP cycles and with the addition of a promoter of hydrogen absorption (thiourea). The thiourea affects the total area and density of pits most significantly.

Does this mean that the localized corrosion is increased by the duration of the c. pol. cycle or only when it's duration is increased in the presence of thiourea. The authors should also clearly state that the thiourea increases the total area density, a great reduction in corrosion would also mean a significant effect. The authors have not made any attempt to improve this issue.

Thank you very much. The conclusion was re-phrased:

The intensity of localized corrosion of pipeline steel X70 increases with an increase of duration of the cathodic period of sign-alternating cycling polarization. This effect was found either with or without the addition of a promoter of hydrogen absorption (thiourea) to background electrolyte. The thiourea increases the total area and density of pits significantly. Thus, effective hydrogen charging of steel during the cathodic period accelerates localized dissolution during the anodic period of SACP.

To clarify these points two phrases were added to the text.

“The same setup of SACP was used either with or without addition of tiourea.”

“Figure 2 shows that the increase of the duration of cathodic polarization increases the corrosion of the steel with or without the addition of thiourea in borate electrolyte. It was determined that in all cases thiourea accelerates the corrosion of the steel comparing with reference electrolyte.”

The main issue is still the overall quality of presentation.

Due to the work and comments of the reviewers, the quality of the presentation was improved. We hope that now the manuscript is more understandable.

 The "non-metal inclusion" should be changed to non-metallic inclusion as is the proper name. 

It was done. Thank you.

Figure quality has improved, but the descriptions under figures should be self explanatory and should make sense on their own, the description under figure 3 "Points 2 - see explanation in the text" is not acceptable. just write non-buffered 0.1 M NaCl, the condition of the testing should also be written under the figures, or in the graph, there is a lot of space, since there is only one curve.

Thank you for comments. The captions to the Figures were rewritten and conditions of the particular experiments were added.

Reviewer 4 Report

The revised paper titled “Effect of sign-alternating cyclic polarization and hydrogen uptake on the localized corrosion of X70 pipeline steel in near-neutral solutions” answered all the reviewer's comments and revised according to the reviewer's comments. The answers and the revised parts seem reasonable, I recommend that this manuscript be accepted after minor revision.

Author Response

The revised paper titled “Effect of sign-alternating cyclic polarization and hydrogen uptake on the localized corrosion of X70 pipeline steel in near-neutral solutions” answered all the reviewer's comments and revised according to the reviewer's comments. The answers and the revised parts seem reasonable, I recommend that this manuscript be accepted after minor revision.

Thank you very much, additional revision of the article was carried out.

Round 3

Reviewer 3 Report

I feel the authors have corrected the manuscript.